# Regulation of Adipose-Derived Stem Cell Activity by Melatonin Receptors in Terms of Viability and Osteogenic Differentiation

**DOI:** 10.3390/ph16091236

**Published:** 2023-09-01

**Authors:** Aleksandra Skubis-Sikora, Bartosz Sikora, Weronika Małysiak, Patrycja Wieczorek, Piotr Czekaj

**Affiliations:** Department of Cytophysiology, Chair of Histology and Embryology, Faculty of Medical Sciences in Katowice, Medical University of Silesia, 40-055 Katowice, Poland

**Keywords:** mesenchymal stem cells, melatonin, regenerative medicine

## Abstract

Melatonin is a hormone secreted mainly by the pineal gland and acts through the Mel1A and Mel1B receptors. Among other actions, melatonin significantly increases osteogenesis during bone regeneration. Human adipose-derived mesenchymal stem cells (ADSCs) are also known to have the potential to differentiate into osteoblast-like cells; however, inefficient culturing due to the loss of properties over time or low cell survival rates on scaffolds is a limitation. Improving the process of ADSC expansion in vitro is crucial for its further successful use in bone regeneration. This study aimed to assess the effect of melatonin on ADSC characteristics, including osteogenicity. We assessed ADSC viability at different melatonin concentrations as well as the effect on its receptor inhibitors (luzindole or 4-P-PDOT). Moreover, we analyzed the ADSC phenotype, apoptosis, cell cycle, and expression of *MTNR1A* and *MTNR1B* receptors, and its potential for osteogenic differentiation. We found that ADSCs treated with melatonin at a concentration of 100 µM had a higher viability compared to those treated at higher melatonin concentrations. Melatonin did not change the phenotype of ADSCs or induce apoptosis and it promoted the activity of some osteogenesis-related genes. We concluded that melatonin is safe, non-toxic to normal ADSCs in vitro, and can be used in regenerative medicine at low doses (100 μM) to improve cell viability without negatively affecting the osteogenic potential of these cells.

## 1. Introduction

In 2023, about 4000 new cases of osteosarcoma were diagnosed. Most of them affect children and teens. Every year 2000 people die from different types of bone cancer. Data show an average 59% survival after diagnosis and 5-year relative survival rate depending on the stage, localization, and type of bone cancer. There are few known risk factors for cancer development and its recurrence. The current treatment focuses on traditional therapies like surgery, chemotherapy, radiotherapy, samarium therapy, and targeted therapy. After treatment, somatic cells and stem cells play essential roles in organism convalescence. Mesenchymal stem cells are crucial for bone regeneration [1,2]. Bone tissue disorders also affect children diagnosed with osteogenesis imperfecta (OI). This genetic disorder is associated with a failure of connective tissue development, with a wide range of phenotypes manifested by a fragile skeleton. There are five known types of OI, which differ mainly in the severity of symptoms. Mild phenotypes offer the chance for treatment to prevent fractures or increase bone mass [3]. Presumably, appropriately primed mesenchymal stem cells, such as adipose-derived stem cells, might be an option for the treatment of OI.

Currently, the reconstruction of bone tissue is mostly focused on autologous or allogenic bone grafting. In the case of obstacles excluding the patient from the surgical grafting procedure, such as a medical condition, difficult-to-access healthy tissue when the surgery requires a large portion of bone, or the lack of a donor, the use of synthetic materials for bone replacement is considered. Any material intended for transplantation must be biocompatible with the human body and possess properties that improve bone reconstruction, such as osteoinduction, osteogenesis, and osteoconduction [4,5]. Many studies have attempted to develop synthetic scaffolds for bone and cartilage reconstruction that are settled with cells aimed at rebuilding bone tissue, such mesenchymal stem cells (MSCs) or induced pluripotent stem cells (iPSCs) [5,6,7,8]. Suitable cell-rich synthetic scaffolds combined with 3D printing technology could successfully replace natural bone. However, even a perfect tissue-equivalent biomaterial will not suffice for treatment without the presence of cells capable of regenerating and rebuilding bone. Most of the cells under investigation for bone reconstruction are MSCs [9], but the usefulness of iPSCs in this area is growing due to their superiority over MSCs in cell proliferation and accessibility. Currently, the potential use of iPSCs in bone regeneration is investigated after their differentiation into MSCs (iPSC-MSCs), and iPSCs are considered an alternative source of MSCs [10,11].

The relatively easy access to MSCs and their natural capacity to differentiate into chondro- and osteoprogenitors make this kind of cell one of the best components for scaffold-based bone reconstruction. Moreover, MSCs can be harvested from patients in a single procedure, while iPSCs can only be obtained after the prior use of genetic engineering techniques during cell reprogramming procedures, which cannot be bypassed and carry additional risks to the patient’s health [12,13].

MSCs, such as bone marrow stem cells or adipose-derived stem cells (ADSCs), are administered to patients by direct injection and are used in bone and cartilage regeneration therapies without the support of biomaterials [14,15]. However, the use of MSCs in regenerative medicine has limitations. The regeneration of bone tissue with stem cells requires a cell population with a high percentage of active cells with differentiation potential. So far, numerous factors have been identified to stimulate stem cell activity and help maintain their stemness, inhibit apoptosis, and improve differentiation capacity [16]. Preclinical studies have shown the positive effects of melatonin on stem cell cultures, especially bone marrow stem cells [16,17].

Melatonin is a biological compound produced by mammalian cells, secreted mainly by the pineal gland [18], but also by the retina, gastrointestinal tract, and lymphocytes [19,20]. It is an agonist of two G-protein-coupled receptors (GPCRs): melatonin receptor 1 type 1A (MEL1A receptor or MT1) coded by the *MTNR1A* gene, and melatonin receptor 2 (MEL1B receptor or MT2) coded by the *MTNR1B* gene. Melatonin receptors are located in the brain [21], but have also been shown to be expressed in adipose tissue [22], human amniotic mesenchymal stem cells [23], and bone marrow mesenchymal stem cells [24]. Melatonin is mainly responsible for synchronizing the circadian rhythm [25], but a number of studies have demonstrated its anti-tumor effects, such as its promotion of apoptosis and antioxidant activity through reactive oxygen species (ROS) inhibition [26] in various types of cancer, e.g., human colorectal cancer HT-29 cells [27], head and neck squamous cell carcinoma (HNSCC) cell lines [28], lung adenocarcinoma cells [29], and ovarian cancer cells [30]. Therefore, melatonin could serve as a supplementary anticancer agent in biomaterial scaffolds or cell cultures for bone reconstruction after major trauma and in operations to remove cancerous tissue, such as in limb-sparing surgery [31]. In addition, high levels of ROS are also generated during chemotherapy and radiation therapy. Melatonin could prevent the negative effects of chemotherapeutics on cells. Some studies have shown that cisplatin-induced toxicity was inhibited by melatonin in cancer cell lines, e.g., human lung adenocarcinoma cells [29], but other studies have proved that melatonin promotes apoptosis and acts on cancer cells synergistically with chemotherapeutics, such as cisplatin [30].

Many compounds can cause undesirable effects on stem cells in vitro, e.g., by disrupting their proliferation and self-renewal potential, modifying their phenotype by inducing spontaneous differentiation into different cell types or, conversely, enhancing differentiation into an undesirable type of cell. The effect of melatonin on healthy normal cells, such as ADSCs, has not been broadly investigated. Many studies have analyzed the use of melatonin in cancer therapy [27,30,32], but there are no results showing the effect of melatonin on ADSC activity, such as apoptosis and differentiation into osteoblast-like cells. This report provides a preliminary investigation into the potential use of melatonin in ADSC preconditioning for therapeutic purposes.

The aim of this study was to evaluate whether melatonin improves the performance of ADSCs in terms of viability and osteogenic differentiation potential. In addition, our goal was to evaluate the possible harmful effects of melatonin by investigating its effect on the induction of apoptosis, and to determine if the mechanism of action is associated with melatonin receptor dependent or independent pathway.

## 2. Results

### 2.1. Effect of Different Melatonin Concentrations on ADSC Viability

The number of living cells was determined using an MTT assay, which consists of the detection of increasing mitochondrial activity in relation to an increased number of proliferating cells. The viability of ADSCs was highest in cultures treated with a 100 μM concentration of melatonin (mean = 187.8% ± 26.7). Our observations showed a significantly lower viability of cells treated with 1000 μM (mean = 13.8% ± 4.6) when compared to untreated CTRL cells (mean = 100%) and to cells treated with 0.1 μM (mean = 121.1% ± 31.9), 1 μM (mean = 151.7% ± 19.8), 10 μM (mean = 165.1% ± 40.4), and 100 μM (*p =* 0.0001) of melatonin. Moreover, the cells treated with 1 μM, 10 μM, and 100 μM (*p =* 0.0001), but not with 0.1 μM, demonstrated a significantly higher viability than the CTRL (Figure 1A, ANOVA, post hoc Tukey, *p* < 0.05, n = 6). A non-toxic 100 µM concentration of melatonin was chosen for further stages of the experiment.

### 2.2. Effect of Melatonin and Its Inhibitors on ADSC Viability

After 48 h of incubation with melatonin and its inhibitors (luzindole and 4-P-PDOT), the bioluminescent ATP assay showed that the viability of ADSCs was highest in the cells treated with melatonin at a concentration of 100 μM (mean = 114.8% ± 5.06). It was significantly higher as compared to the untreated cells (CTRL) (mean = 100%) and the cells treated with melatonin with luzindole (MEL + LUZ, mean = 103.4% ± 6.6), melatonin with 4-P-PDOT (MEL + 4-P-PDOT, mean = 79.1% ± 12.9), luzindole (mean = 105.4% ± 2.6), and 4-P-PDOT (mean = 100.8% ± 5.3) (*p =* 0.0001) (Figure 1B).

Both the MTT and ATP assays showed that the addition of 100 µM of melatonin improved ADSC viability. The ATP assay also showed that melatonin receptor inhibitors are not toxic to ADSCs, which was relevant for further procedures.

### 2.3. Morphology and Immunophenotype of ADSCs after Treatment with Melatonin and Its Inhibitors

Unstained ADSCs were microscopically examined (under a bright field) after 48 h of incubation with melatonin and its inhibitors. Any significant changes in general morphology, vacuolization, detachment, cell lysis, and membrane integrity were investigated after a 48 h cell culture. The ADSCs retained a fibroblast-like shape and adhered to the plate (Figure 2).

The surface markers for the mesenchymal stem cells CD73, CD90, and CD105 were identified on the ADSCs 48 h after cell incubation with melatonin (MEL), melatonin with luzindole (MEL + LUZ), melatonin with 4-P-PDOT (MEL + 4-P-PDOT), and the standard medium (CTRL). No statistically significant differences were observed between the analyzed cultures in the number of positive cells for each of the visualized markers (Figure 3, ANOVA, post hoc Tukey, *p* < 0.05, n = 3). Over 90% of the cells expressed CD73, CD90, and CD105 in all the examined groups, and melatonin did not change the cell phenotype or morphology.

### 2.4. Effects of Melatonin on ADSC Apoptosis and Necrosis

Apoptosis was assessed using an Annexin/IP assay with FACS in all the examined groups (ANOVA, post hoc Tuckey, *p* < 0.05, n = 3). This assay analyzed the percentage of live-, early-, or late-apoptotic and necrotic cells within the experimental groups.

In all the investigated cultures, the percentage of live cells (mean = 91% ± 1.87) was significantly higher compared to the apoptotic or necrotic cells, and was comparable among the experimental groups (Figure 4). At the same time, there was an average of 3.16% ± 0.56 early-apoptotic and 5.61% ± 1.8 late-apoptotic cells and a very small percentage of necrotic cells (mean = 0.13% ± 0.02). This indicates that melatonin does not induce apoptosis in ADSCs.

Furthermore, cleaved caspase-3 was detected with immunofluorescence to visualize the apoptosis (Figure 5). The activity of cleaved caspase-3 in the ADSC cultures was very weak or absent in both the standard medium (CTRL) and in the media supplemented with melatonin and its inhibitors. It confirmed the very weak apoptotic activity assayed using FACS in all the investigated cell cultures.

### 2.5. Effects of Melatonin on Cell Cycle in ADSCs

The cell cycle was assessed using a flow cytometric evaluation of the size of the populations in the G1/G0, S, or G2/M phase of the cycle (Figure 6). The ADSCs cultured with melatonin and its inhibitors showed a similar distribution of cell-cycle phases compared to the untreated (CTRL) cells. The highest percentage of cells was observed in the G1/G0 phase (mean = 89% ± 2.96). The cells in the S and G2/M phases constituted only a minor percentage, 2.7% ± 0.65 and 8.4% ± 2.43, respectively. This suggests the sustained proliferative activity of ADSCs treated with melatonin and its inhibitors. We did not observe significant differences between the groups in the G2/M phase specific for the growth and division of cells; it proved that melatonin did not change cell division activity (ANOVA, post hoc Tukey, *p* < 0.05).

### 2.6. Effects of Melatonin on Caspases (CASP3, CASP7) and Melatonin Receptors (MTNR1A, MTNR1B) Gene Expression

After culturing the ADSCs with melatonin and its inhibitors, the RNA was isolated and the gene expression was assayed. No apoptosis-specific gene expression (*CASP3* and *CASP7*) was detected at the mRNA level, which suggests that apoptosis-related genes were not activated. The expression of *MTNR1A* and *MTNR1B* genes were detected in all the examined groups (Figure 7). The expression of *MTNR1A* and *MTNR1B* in the untreated cells (CTRL) and the cells treated with melatonin (MEL) were higher than in the cells incubated with melatonin and luzindole (MEL + LUZ), but the differences were not statistically significant (Kruskal–Wallis test with multiple comparisons, *p* < 0.05). A similar relationship was observed between the CTRL and MEL groups and MEL + 4-P-PDOT; however, in the latter case, the differences were statistically significant (*p <* 0.05). This confirms that melatonin receptors are present on ADSCs and their lowered expression on cells treated with melatonin inhibitors suggests that the inhibition was effective.

### 2.7. Effect of Melatonin on Hydroxyapatite and Calcium Deposition after Cell Differentiation

After 7 days of differentiation of the ADSCs with the osteogenic medium, the extracellular calcium deposition and hydroxyapatite identification were assessed. The results were comparable among the experimental groups when compared to ADSCs after osteogenic differentiation (CTRL_O) without the addition of melatonin, luzindole, or 4P-PDOT, and to undifferentiated ADSCs (CTRL neg) (Figure 8A). The quantified level of hydroxyapatite was significantly lower in the treated and CTRL neg cells (Figure 8B, ANOVA, post hoc Tukey, *p* < 0.05). The data show that mineralization appeared in all the experimental groups. However, the total amount of hydroxyapatite was lower in the treated cells, which suggests that ADSC pretreatment with melatonin can slow down the mineralization process but does not inhibit it. This effect may also be associated with an increased cell proliferation of osteoblasts, which appears prior to mineralization.

### 2.8. Effect of Melatonin on the Expression of Osteogenesis-Related Genes BGLAP, SPP1, and RUNX2

The effect of melatonin on the expression of osteogenesis-related genes was assessed after 7 days of ADSC differentiation. The expression of *BGLAP, SPP1,* and *RUNX2* genes in the cells treated with melatonin was higher as compared to untreated controls, but only in the case of *SPP1* was it statistically significant (Figure 9, ANOVA, post hoc Tukey, *p* < 0.05). Moreover, we observed a higher expression of *RUNX2* and *SPP1* in the ADSCs treated with melatonin compared to the cells incubated with luzindole and 4P-PDOT. There were no statistically significant differences in the expression of *BGLAP* among the examined groups: CTRL_O, MEL + LUZ_O, and MEL + 4-P-PDOT. This result suggests that melatonin can increase the expression of some of the examined genes. It cannot be excluded that the expression of *BGLAP* and *RUNX2* could be shifted in time compared to *SPP1*.

### 2.9. RUNX2 and Osteocalcin Protein Concentrations in Differentiating ADSCs

The concentration of RUNX2 protein, a critical transcription factor associated with differentiation into osteoblast-like cells, was assessed in the ADSCs after a 7-day differentiation (Figure 10, ANOVA, post hoc Tukey, *p* < 0.05). Our observations showed a higher concentration of RUNX2 in the cells treated with melatonin compared to the cells incubated with melatonin + luzindole or 4-P-PDOT. The levels of RUNX2 were also higher in the melatonin group versus the CTRL, but with no statistical significance. The concentration of RUNX2 was consistent with the expression of the *RUNX2* gene.

Osteocalcin (OCN) regulates the processes of calcium binding and the deposition of minerals. The concentration of the OCN protein specific for fully functional osteoblasts was assessed after a 7-day differentiation. A slightly lower OCN expression was observed in the cells treated with melatonin and melatonin with 4-P-PDOT, as compared to the control cells. Lower levels were observed in the cells incubated with melatonin with 4-P-PDOT as compared to melatonin with luzindole (*p <* 0.05). The observation that the concentration of OCN was lower in the MEL_O group is similar to the changes noted in the level of hydroxyapatite during mineralization. These changes could be associated with the increased proliferation of osteoblasts or the early time point of measurement, but they indicate that osteogenesis is still active in the ADSCs treated with melatonin.

### 2.10. RUNX2 and Osteocalcin Immunodetection in Differentiating ADSCs

The RUNX2 and osteocalcin proteins were visualized with immunofluorescence. The differentiated ADSCs expressed both proteins in all the examined groups, including the untreated cells (CTRL) (Figure 11).

## 3. Discussion

Clinical studies have shown that stem cells can be useful in the treatment of bone cancers [33] as an adjuvant therapy before the introduction of drugs. Other studies have shown that melatonin may be useful in anti-cancer therapy and, when combined with synthetic scaffolds, may promote osteogenesis and bone regeneration [26,27,28,29,30,32]. These are promising findings for developing new therapy strategies for osteosarcoma, which still does not have a satisfactory treatment. Around 40 out of 100 people survive this disease more than 5 years after diagnosis. Effective chemotherapy with limb-sparing surgery, which involves removing a section of the affected bone with a surrounding healthy tissue margin and replacing it with a graft, is nowadays the best and most-used treatment option. Limb-sparing surgery provides patients with a better quality of life than amputation. Grafts can be taken from the human body, but sometimes the bone loss is too severe. In this case, the use of synthetic biomaterials seeded with stem cells is considered as an alternative method of tissue replacement [4,6,33]. We assume that in the future, combining stem cells and melatonin with biomaterial-based scaffolds that could control its release will improve bone regeneration after the resection of fragments affected by osteosarcoma.

Searching for new strategies for bone regeneration is crucial not just for bone cancer treatment. A serious medical condition related to bone formation disorders is the genetically determined osteogenesis imperfecta (OI). This genetic disorder is characterized by a highly decreased bone mass leading to multiple fractures associated with a fragile skeleton. To date, it has been reported that melatonin has anti-osteoporotic and pro-osteogenic activity [34,35]. Several clinical trials have been carried out which have recognized the potential usefulness of mesenchymal stem cells (MSCs) in OI treatment or in other bone disorder-related conditions (NCT02172885, NCT04623606, NCT05559801, NCT03706482, NCT00187018, NCT00705120, and NCT00186914). Some of the reported results from preclinical and clinical studies in this field provide promising evidence of the effects of MSCs [36,37]. However, in the absence of family member as a close kinship donor, or impossible or difficult access to autologous cells, the need for investigating new stem cell sources and their properties’ improvement is essential. Priming ADSCs in vitro could be possibly beneficial for OI therapy and could result in a better outcome [37]. Moreover, data has shown that osteoporosis is promoted by oxidative stress [38]. Melatonin treatment might inhibit ROS production and improve osteogenesis by acting on stem cells and inhibiting osteoporosis by reducing oxidative stress. The pre-treating of ADSCs with melatonin could improve their viability and osteogenic potential. Such primed ADSCs probably will bring better outcomes in therapy.

The relatively few and limited reports on melatonin’s effects on mesenchymal stem cells prompted us to investigate the direct influence of it on ADSCs and their osteogenic potential. Melatonin is known to act through receptor or non-receptor pathways [24,39]. Our report partly shows that melatonin’s effect on ADSCs is presumably exhibited through both pathways, and is dependent on the investigated process’s specificity. Starting our experiments, we assessed if the examined compounds altered the cell viability measured by the ATP level, reflecting the mitochondrial activity which takes place only in living cells. Our experiments investigated the influence of the potential usefulness of melatonin in ADSC culturing and focused on improving cell conditions in the culture dish. According to our hypothesis, melatonin was expected to improve ADSC viability, which we confirmed with an ATP assay. The data presented in Figure 1B indicate that melatonin significantly improved ADSC activity; furthermore, this assay also verified the potential harmful effects of the remaining compounds tested in this study. We tested melatonin (MEL) and receptor antagonists, such as luzindole (LUZ), which is a competitive, non-selective melatonin receptor antagonist of both melatonin receptor type 1A and 1B, and 4-phenyl-2-propionamidotetralin (4-P-PDOT), which is a selective antagonist of melatonin receptor type 1B [40]. The remaining study groups consisted of combinations of MEL and LUZ or MEL and 4-P-PDOT used simultaneously in treating ADSC, to assess what pathway the melatonin acted through. However, whether the melatonin receptors are expressed in ADSCs has not yet been directly reported. Regarding this question, we conducted an evaluation of the mRNA expression coding *MTNR1A* and *MTNR1B*. Our findings confirmed that both genes are expressed in ADSCs. Also, we investigated melatonin’s effect on their expression by treating ADSCs only with melatonin (MEL) or with the addition of melatonin and its receptor antagonists (LUZ or 4-P-PDOT). Surprisingly, the comparative analysis showed that the level of mRNA in the MEL group is similar to that of untreated cells (CTRL), but is lower with MEL + 4-P-PDOT, where the difference was statistically significant. An emphatically lowered mRNA level was also observed in the MEL + LUZ group; however, it was statistically irrelevant. In our opinion, this finding confirms that LUZ or 4-P-PDOT successfully acts on melatonin receptors and inhibits their activity, as expected. In turn, melatonin does not disturb their expression, which suggests that it probably acts through non-receptor pathways. This finding was supported by our further assessment. We aimed to check if melatonin somehow disturbs the mineralization of ADSC during osteogenesis. Cells were cultured in an osteogenic medium with the addition of melatonin or in combination with its receptor antagonists. We also provided a negative control of an osteogenic medium. Interestingly, the mineralization measured using the concentration of hydroxyapatite (RFU) showed that the total amount of this osteogenic marker is significantly lowered in the MEL, MEL + LUZ, and MEL + 4-P-PDOT groups compared to the CTRL. As the melatonin antagonists are expected to revert melatonin’s effect on mineralization, we assume that this confirms that melatonin acts on ADSCs through the non-receptor pathway in this case.

One of the issues associated with ADSC growth in vitro is the loss of its properties and activity over time. The decreasing activity of ADSCs in vitro leads to an inefficient culture and low cell survival on scaffolds [16]. It is essential to ensure the best possible culture conditions to maintain high levels of specific surface markers, whose pattern must meet the requirements defined by the International Society for Cell and Gene Therapy (ISCT) [19]. Unfortunately, multiple passages and long-term culturing of ADSCs are known to alter their phenotype and properties [17]. In this study, we demonstrated the effect of melatonin on normal, healthy ADSCs taken for the experiments at the third passage. They were characterized by a high mitochondrial activity and osteogenic differentiation ability. These characteristics met the standards for cells that could be used in clinical practice [41].

It should be kept in mind that the factors influencing the effect of melatonin on cells in vitro are the duration and mode of exposure, namely, continuous incubation or pre-treatment [24,26,42]. Studies have shown various effects of melatonin after 24 h of treatment, such as an enhancement of viability and anti-apoptotic and anti-oxidative activity [16]. We assumed that extending the duration of treatment to 48 h was more appropriate for analyzing its influence on stem cell activity after the potential incorporation of melatonin into the transplanted biomaterial scaffold.

Another important point is that the effect of melatonin can be both dose and cell-type dependent. Melatonin at a physiological concentration of 50 nM (0.05 μM) can regulate many processes [43], but concentrations higher than the physiological concentration produced better effects [24]. Rubio et al. noted that melatonin at a concentration of 1000 µM (1 mM) significantly reduced the mitochondrial activity of cells [44]. We assessed a toxic dose of melatonin on ADSCs and demonstrated a distinct difference in the effects induced at concentrations of 100 µM (0.1 mM)—which had the highest cell viability, and at 1000 µM (1 mM)—which exhibited decreased viability and the death of the cells. By measuring the viability and using specific receptor inhibitors, we confirmed that melatonin at a dose of 100 µM enhances the activity of ADSCs in vitro. Moreover, melatonin and the inhibitors of its receptors did not influence the morphology of the examined cells. A subsequent evaluation of the effect of melatonin on the cell cycle showed that a 48 h incubation of the cells with melatonin at a concentration of 100 μM did not induce cell death, and showed a similar distribution of relative cell-cycle phases as in the untreated control. Also, Plaimee et al. showed that untreated cells and cells treated with melatonin are similar and represent the G0/G1 phase [29]. Chan et al. observed that melatonin at a concentration of 100 µM promotes the proliferation of MSCs isolated from dental pulp and does not affect cell morphology [45]. On the other hand, Bejarano et al. showed that a higher concentration (1000 μM) and longer incubation (72 h) with melatonin induces the transition of cells into the sub-G1 phase and promotes apoptosis [32]. In our examined populations, we did not observe any cells in the sub-G1 phase after treatment with melatonin and its antagonists, indicating that the cycle was not arrested by melatonin.

Our further observations of the ADSC phenotype for the potential cytotoxicity of melatonin and its inhibitors indicated that after a 48 h incubation, over 95% of the ADSC population expressed the surface proteins CD73, CD90, and CD105, which are characteristic of MSCs [46]. The lack of differences between the examined groups confirms the results obtained by Chan et al. [45], who observed that over 93% of the MSCs that were incubated for 24 h with 100 μM melatonin were still positive for CD73, CD90, and CD105 markers. This proves that melatonin does not influence the immunophenotype of ADSCs.

Another important point in terms of the practical application of melatonin in bone regeneration is that melatonin acts pro- or anti-apoptotically, depending on the type of cell, tissue, and pathology [47]. Moreover, there are no similar reports describing the synergistic use of MSCs and melatonin in, e.g., osteosarcomas. Some other studies have proven that melatonin in a hypoxic environment inhibits pro-apoptotic proteins and caspase-3 [48] and, moreover, inhibits proliferation in cancer cells, like prostate [49] and melanoma [50], through the melatonin receptor pathway. It has also been shown that melatonin at a concentration of 5 μM is able to protect MSCs from apoptosis and ROS activity [16]. In this study, we analyzed the effect of melatonin (100 μM, 48 h) on the apoptosis of healthy ADSCs, which could potentially be used to regenerate damaged bone. We did not observe the enhanced expression of cleaved caspase-3 at either the mRNA or protein levels. Using flow cytometry, we showed that 90% of the cells treated with melatonin were still alive, and only an average of 3.16% and 5.61% cells were at the early- and late-apoptosis phases, respectively. The remaining, very small population consisted of necrotic cells. However, a high melatonin concentration (1 mM) promoted higher caspase-3 activity after 24–72 h of incubation, inhibiting human myeloid leukemia cells and activating apoptosis by non-receptor pathways [44]. Also, Bejarano et al. showed that 1 mM of melatonin induces caspase-3, reaching the maximal effect at 12 h of incubation and then decreasing 48 h after cell treatment [32]. Both of the above studies examined melatonin concentrations that were 10-fold higher than those we used in our experiment.

Finally, we analyzed the effect of melatonin on the differentiation of ADSCs into osteoblast-like cells by adding melatonin and/or its inhibitors to an osteogenic medium and culturing the cells for 7 days. The ADSCs produced calcium deposition and hydroxyapatite in all the differentiation groups, but we observed that hydroxyapatite production was lower in the melatonin and luzindole/4-P-PDOT groups. Moreover, the standard osteogenic medium induced denser calcium deposition. These results are similar to those of Radio et al., who showed that 0.05 µM melatonin increased ALP activity specific for osteoblasts through receptor-dependent pathways, which was associated with the activation of MEK/ERK proteins. Rafat et al. also showed that preconditioning the cells with melatonin improves osteogenic differentiation [23,51]. The osteogenesis of mesenchymal stem cells consists of proliferation, differentiation, and mineralization. Here, we observed an effect related to an early phase of this process. This suggests that an extended differentiation time may yield more specific results about the effect of melatonin on this process. Several studies have proven that the mineralization of osteoblast-like cells on scaffolds can be more challenging than the efficient culturing of stem cells [52]. Therefore, our results prove that melatonin does not inhibit ADSCs-dependent osteogenesis during a 7-day differentiation.

Moreover, the expression of genes related to early (*SPP1*, *RUNX2*) and late (*BGLAP*) osteogenesis was detected in all the groups. In the melatonin-treated cells, we observed a higher expression of *SPP1* mRNA encoding the glyco-phosphoprotein osteopontin, regulated by *RUNX2* in the early phase of osteogenesis, together with osteocalcin [44,45], suggesting better osteogenic activity in this group. In addition, *RUNX2*, a mediator of ADSCs differentiation into osteoblasts [53], was also overexpressed in the melatonin-treated cells as compared to the groups treated with melatonin and its inhibitors. The same relationships were observed at the protein level. This may indicate that the process of ADSCs differentiation is related to melatonin receptors, which can regulate the expression of RUNX2. The addition of luzindole or 4-P-PDOT in combination with melatonin into the ADSC cell culture decreased the expression of *RUNX2* and *SPP1* genes as compared to the cells treated with melatonin alone, similarly to the study of Zhang et al., whose data showed melatonin enhanced Runx2 expression [35]. This suggests that melatonin may regulate the expression of early osteogenesis-related genes through receptor pathways.

The expression of *BGLAP* encoding osteocalcin was detectable in all the cultures, regardless of whether or not they were supplemented with melatonin and its inhibitors, with the highest, but not statistically significant, values in the melatonin group. The protein levels identified with ELISA were similar in all the examined groups. Osteocalcin is known to mediate mineralization during osteogenesis, and it appeared uniformly in the cells over several days of differentiation [54]. Higher concentrations of osteocalcin are secreted by mature osteoblasts [55,56]. We analyzed the cells after 7 days of differentiation and, at this time point, the factor associated with the early phase of osteogenesis, RUNX2, was at a high level. It is likely that the period of 7-day differentiation may be too short to detect osteocalcin at a concentration similar to that of RUNX2, a transcription factor that regulates its expression.

In conclusion, melatonin did not change the phenotype of the ADSCs or induce apoptosis, and it promoted the activity of some osteogenesis-related genes. Melatonin seems to be safe, non-toxic to normal ADSCs in vitro, and can be used in low doses (100 μM) to improve cell viability, while not affecting the osteogenic potential of these cells. Melatonin may be an important factor for ADSC preconditioning in cell cultures preceding their use in regenerative medicine. However, this study has limitations. The presented data are lacking an analysis of the influence of melatonin on cells recovering from pathological conditions or that have been affected by drugs, such as chemotherapeutics. Thus, it cannot be concluded how it would act during potential cancer therapy. Also, the effects of melatonin on osteogenesis presented here were tested only after 7 days of differentiation. It is advisable to study this effect at a few time points to assess if osteogenesis is, similarly to our findings, in fact slightly slowed down by melatonin, or if it is improved after an extended time of treating stem cells.

Some studies have proven that melatonin promotes osteogenesis more efficiently in pathological cases than in osteoporotic defects [34,57] or periodontitis [58]. Data have shown that this process is related to the activation of the PI3K/AKT/mTOR pathway in stem cells after cell damage with, e.g., dexamethasone [57]. Presumably, melatonin acts in the same way after the treatment of cells with cytostatic drugs, such as cisplatin or doxorubicin, used in the treatment of osteosarcoma. Our obtained results answered preliminary questions. We showed that melatonin in the examined doses might be useful in ADSC priming prior to its potential application in therapy. Further investigations will focus on its effects on cells subjected to conditions reflecting cancer treatment, like the influence of chemotherapeutics or even radiation. Studying melatonin’s influence on recovering cells will answer whether it is effective in such an extreme environment. Moreover, other melatonin receptor agonists, like tasimelteon, might act similarly to melatonin, which could be considered for study when planning future experiments. The mechanism of melatonin activity should be analyzed more precisely. Further studies will focus on better understating its observed effects. For example, it is possible that PI3K/AKT/mTOR pathway plays a crucial role in melatonin activity, which is also advisable to be investigated for a better understanding of the data presented in this report [59].

## 4. Materials and Methods

### 4.1. Experiments

The experiments were conducted on commercially acquired normal human adipose-derived mesenchymal stem cells (PCS-500-011, ATCC, Manassas, VA, USA) at the biosafety level 2 laboratory (BSL2) in our department. No additional approval of the local ethical committee was needed.

The effect of melatonin was analyzed in several aspects. First, ADSC viability was evaluated under the influence of melatonin at the following concentrations: 0.1 µM, 1 µM, 10 µM, 100 µM, and 1000 µM, and compared to untreated control cells (CTRL). A selected non-toxic concentration of melatonin was used for further proceedings.

Next, cells were incubated for 48 h with 100 µM melatonin and with/without inhibitors: 10 µM of luzindole (N-acetyl-2-benzyltryptamine, L-2407-5mg, Sigma Aldrich, Saint Louis, MO, USA), which is an antagonist of MT1 and MT2 receptors, or 10 µM of 4-P-PDOT (4-phenyl-2-propionamidotetralin, SML1189-5MG, Sigma Aldrich, USA), the selective antagonist of the MT2 receptor. Cells were assessed based on ATP activity. The effects of melatonin (MEL), melatonin with luzindole (MEL + LUZ), melatonin with 4-P-PDOT (MEL + 4-P-PDOT), luzindole without melatonin (LUZ), and 4-P-PDOT without melatonin (4-P-PDOT) were compared.

The changes in cell phenotype were then analyzed with FACS using the surface markers CD73, CD90, and CD105. The cell cycle was evaluated with propidium iodide (PI). Apoptosis was assessed using an Annexin V assay with FACS, immunofluorescence staining of cleaved caspase-3, and expression of the apoptosis-related genes *CASP3* and *CASP7* after a 48 h incubation with melatonin and its inhibitors. Moreover, the expression of *MTNR1A* and *MTNR1B* receptor genes was assessed with RT-qPCR.

Finally, the ability of ADSCs to differentiate into osteoblast-like cells after treatment with/without melatonin and its inhibitors was assessed. ADSCs were treated with MEL, MEL + LUZ, and MEL + 4-P-PDOT for 7 days and cultured in an osteogenic medium. After osteo-differentiation, cells were stained for the detection of calcium deposits and hydroxyapatite. Moreover, expression of the osteogenesis-related genes runt-related transcription factor 2 (*RUNX2*), bone gamma-carboxyglutamate protein (*BGLAP*), and secreted phosphoprotein 1 (*SPP1*) were analyzed with RT-qPCR, while the level of osteoblast-specific proteins RUNX2 and osteocalcin were analyzed with an enzyme-linked immunosorbent (ELISA) and immunofluorescence.

### 4.2. Cell Culture Conditions

ADSCs were maintained in DMEM/F-12 (11320033, Dulbecco’s Modified Eagle Medium/Nutrient Mixture F-12, Gibco by Thermo Fisher Scientific, Waltham, MA, USA), supplemented with fetal bovine serum (ECS0180L, 10%, FBS, EuroClone, Italy), and a penicillin–streptomycin mixture (09-757F, 1%, Lonza, Basel, Switzerland) at 37 °C in a 5% CO_2_ incubator (Sanyo MCO-19M, Osaka, Japan). The culture medium was changed every two days.

The experiment was performed on cells in the logarithmic phase of growth under the condition of ≥90% viability. ADSCs used for the experiment were at the third passage. Cells were assessed using an Olympus IX73 microscope (Olympus, Shinjuku, Tokyo, Japan), which was used for photographic documentation.

### 4.3. Cell Viability

#### 4.3.1. MTT Assay

ADSCs were plated at a density of 3 × 10^3^ per well in 96-well plates and treated with different concentrations of melatonin (M5250-1G, Sigma Aldrich): 0.1 µM, 1 µM, 10 µM, 100 µM, and 1000 µM. After a 48 h incubation with melatonin and its inhibitors, the number of viable cells was evaluated using an MTT assay (M2128, Thiazolyl Blue Tetrazolium Bromide, Sigma-Aldrich, USA) according to the manufacturer’s protocol. Next, MTT solution was added, and cells were incubated for 3 h at 37 °C in 5% CO_2_ using a Sanyo MCO-19M incubator (Sanyo, Japan). The medium was then aspirated and DMSO was added to cells for 1 h to dissolve the formazan crystals. Cell proliferation assay was performed in eight replicates (n = 8). Untreated cells served as the control (CTRL), whereas cells treated with 1% Triton X (93443, Sigma-Aldrich, USA) served as the negative control (CTRL neg). Finally, the absorbance of formazan was measured at a wavelength of 570 nm using a VICTOR Nivo microplate reader (PerkinElmer, Waltham, MA, USA).

#### 4.3.2. Quantifying ATP

Cells were plated at a density of 3 × 10^3^ per well in 96-well black plates. The effect of melatonin (MEL), melatonin with luzindole (MEL + LUZ), melatonin with 4-P-PDOT (MEL + 4-P-PDOT), luzindole without melatonin (LUZ), and 4-P-PDOT without melatonin (4-P-PDOT), compared to that of untreated cells (CTRL), were analyzed using the CellTiter-Glo 2.0 Assay kit (G9241, Promega, USA) according to the manufacturer’s protocol. The bioluminescent CellTiter-Glo 2.0 Assay determines the number of viable, metabolically active cells in cultures by quantifying ATP. It is based on a luciferase enzymatic reaction and uses ATP from viable cells to generate photons of light. After a 48 h incubation, the plate with cells and CellTiter-Glo 2.0 reagent were equilibrated to RT for 30 min. Next, a CellTiter-Glo 2.0 Reagent (100 µL) was added to each well of the cell culture plate. Additionally, control wells containing medium without cells were prepared to determine background luminescence. The cell viability assay was performed in eight replicates (n = 8). The contents were then mixed for 2 min on an orbital shaker and incubated at RT for 10 min. Finally, luminescence directly proportional to the number of viable cells in the culture was recorded using a VICTOR Nivo microplate reader (PerkinElmer, USA).

### 4.4. Cell Phenotyping

Flow cytometry analysis (FACS analysis) was used to characterize cells by identifying fluorochrome-conjugated surface proteins CD73-CFS, CD90-APC, and CD105-PerCP, with the use of Human Mesenchymal Stem Cell Verification Flow Kit (FMC020, R&D Systems, USA). The effect of melatonin (MEL), melatonin with luzindole (MEL + LUZ), and melatonin with 4-P-PDOT (MEL + 4-P-PDOT) were analyzed after 48 h of ADSC culture and compared to untreated cells (CTRL).

### 4.5. Cell-Cycle Analysis

FxCycle PI/RNase Staining Solution (F10797, Thermo Fisher Scientific, Waltham, MA, USA) was used for flow cytometric analysis of the cell cycle and recognition of G0/G1, S, and G2/M mitotic phases in cell populations. After a 48 h culture with melatonin (MEL), melatonin with luzindole (MEL + LUZ), melatonin with 4-P-PDOT (MEL + 4-P-PDOT), and standard medium, cells were fixed in 70% ethanol and incubated with staining solution at RT for 30 min. Samples were analyzed at excitation wavelengths of 488 nm and 532 nm in triplicate (n = 3).

### 4.6. Evaluation of Apoptosis

#### 4.6.1. Cytometric Examination

FITC Annexin V Apoptosis Detection Kit I (556547, BD Biosciences, Franklin Lakes, NJ, USA) was used to demonstrate apoptosis in cultured cells using flow cytometry. This allowed us to select four populations of cells: live, early apoptotic, late apoptotic, and dead after 48 h of incubation. All samples were analyzed in triplicate (n = 3).

#### 4.6.2. Cleaved Caspase-3 Immunofluorescence Detection

After 48 h of cell treatment, ADSCs were fixed with 4% formaldehyde for 15 min. Cleaved caspase-3 was determined using first a polyclonal primary Cleaved Caspase-3 (Asp175) antibody (#9661, Cell Signalling Technology, Danvers, MA, USA), and then a fluorochrome-conjugated Alexa Fluor 488 secondary antibody (ab150113, Abcam, Cambridge, UK). Cell nuclei were visualized with DAPI dye (VECTASHIELD Vibrance Antifade Mounting Medium, H-1700, Vector, Newark, CA, USA). Positive controls were adherent ADSCs incubated with 2% DMSO to induce apoptosis. The isotype control was a rabbit monoclonal antibody (ABIN569326, Abcam).

### 4.7. Osteogenic Differentiation of ADSCs

ADSCs were differentiated into osteoblast-like cells using osteogenic medium. Cells were plated and grown until 70% confluence. Subsequently, the medium was replaced with OsteoMAX-XF Differentiation Medium (SCM121, Sigma-Aldrich, USA) and cells were cultured for 7 days. Cells were then fixed for Alizarin red staining and hydroxyapatite (HA) detection and immunofluorescence staining of marker proteins. The cells were collected for the identification of proteins using immunoassay (ELISA) and molecular analysis with RT-qPCR.

#### 4.7.1. Visualization of Calcium Deposits

Alizarin red staining was used to recognize calcium deposits secreted by differentiating cells. Alizarin red dye (A5533, Sigma Aldrich, USA) at a concentration of 40 mM was prepared in ddH_2_O and the pH was adjusted to 4.1 using 10% ammonium hydroxide (NH_4_OH, 221228, Sigma Aldrich, USA). ADSCs were fixed with 4% paraformaldehyde (1.00496.9011, Sigma Aldrich, USA) for 15 min at RT. The fixative was then removed and cells were washed with ddH_2_O. After water removal, cells were stained for 20 min with Alizarin red dye using orbital shaking. Dye was removed and cells were washed 5 times with ddH_2_O. An Olympus IX73 microscope (Olympus, Shinjuku, Tokyo, Japan) was used for photographic documentation.

#### 4.7.2. Hydroxyapatite Detection

Hydroxyapatite (HA, Ca10(PO4)6(OH)2)) was detected with OsteoImage Mineralization Assay (PA1503, Lonza, USA) for evaluation of the mineralization process in cell cultures. After differentiation, the culture medium was removed, cells were washed with PBS (21-040-CV, Corning, Glendale, AZ, USA), fixed with ethanol (396480111, POCH, Gliwice, Poland) and rinsed with wash buffer. Next, the staining reagent was added and cells were incubated in the dark at RT for 30 min and then washed again. Finally, the fluorescence was read at 495 nm in a 96-well plate using a VICTOR Nivo microplate reader (Perkin Elmer, USA). The detected green fluorescent signal was proportional to mineralization. An Olympus IX73 microscope (Olympus, Shinjuku, Tokyo, Japan) was used for photographic documentation.

#### 4.7.3. Immunodetection of Osteocalcin and RUNX2

After ADSC differentiation, the obtained cells were fixed with 4% formaldehyde for 15 min. This was followed by permeabilization with 0.5% Triton X solution (93443, Sigma-Aldrich, USA). The proteins osteocalcin and RUNX2 were detected after incubation of the samples with anti-osteocalcin monoclonal antibody OCG3 (1:500, ab13420, Abcam, UK) and anti-RUNX2 monoclonal antibody 2B9 (1:50, ab76956, Abcam, UK) primary antibodies, respectively. Subsequently, one of two fluorochrome-conjugated secondary antibodies was used: Alexa Fluor 488 (1:1000, ab150113, Abcam, UK) or Alexa Fluor 568 (1:1000, ab175473, Abcam, UK). Cell nuclei were stained with DAPI dye (VECTASHIELD Vibrance Antifade Mounting Medium, H-1700, Vector, USA). The isotype controls were Mouse IgG2a (ab18415, Abcam, UK) and IgG3 (ab18392, Abcam, UK).

#### 4.7.4. Immunoenzymatic Assay of RUNX2 and Osteocalcin

The concentrations of RUNX2 and osteocalcin proteins in the differentiating cells were analyzed with the use of immunoenzymatic tests (Human CBFA1/RUNX2 ELISA Kit LS-F4390 (LSBio, Seattle, WA, USA) and Human Osteocalcin Instant ELISA Kit BMS2020INST (Thermo Fisher Scientific, USA)) according to the manufacturers’ protocols. Cells were lysed using a freeze (−20 °C)/thaw (RT) procedure 5 times. Optical density (OD) was read at 450 nm using a VICTOR Nivo microplate reader (Perkin Elmer, USA). Three technical replicates were performed for each biological replicate.

#### 4.7.5. Expression of Genes Related to Apoptosis and Osteogenesis

Total RNA was extracted from cells using the High Pure RNA Isolation Kit (11828665001, Roche, Indianapolis, IN, USA). RNA concentration was determined using a Nanodrop 2000 spectrophotometer (Thermo Fisher, USA).

Expression of the caspases, *CASP3* and *CASP7*, related to apoptosis, and melatonin receptor genes, *MTNR1A* and *MTNR1B*, were assessed prior to cell differentiation. Expression of the genes *RUNX2*, *BGLAP*, and *SPP1* related to osteogenesis were evaluated in the differentiating cells.

Gene expression was determined using a real time RT-qPCR technique with SYBR Green chemistry (07800177001, LC EvoScript RNA SYBR Green Master, Roche, USA) and LightCycler 96 Instrument (Roche, USA). All samples were tested in triplicate. β-Actin was included as an endogenous positive control (housekeeping gene) of amplification and integrity of the RNA extracts. Oligonucleotide primers were obtained from Sigma-Aldrich company (Sigma-Aldrich, USA). Reactions were completed using melting curve analysis to confirm the specificity of amplification and absence of primer dimers.

### 4.8. Statistical Analysis

Statistical analysis was performed using Statistica 13.3 software (TIBCO, Palo Alto, CA, USA). Shapiro–Wilk test (normality test) was used for assessment of data distribution. Study groups were compared using Kruskal–Wallis test for non-normally distributed data and values were presented as the median (Me), with the 25th and 75th quartiles and minimum and maximum. ANOVA with post hoc Tukey test was used for normally distributed data and values were presented as mean and standard deviation. The level of significance was set at 5% (α = 0.05) for all statistical tests.

## 5. Conclusions

The data presented indicate that melatonin is not toxic to normal ADSCs in vitro at low doses. The dose of 100 µM promotes ADSC viability. Melatonin receptors are expressed in ADSCs. Presumably, the mechanism of action of melatonin on the osteogenesis of ADSCs is exhibited through a receptor-independent pathway. Melatonin does not interfere with the controlled osteogenic differentiation of ADSCs. Luzindole and 4-P-PDOT inhibit the expression of genes encoding melatonin receptors.

## Figures and Tables

**Figure 1 pharmaceuticals-16-01236-f001:**
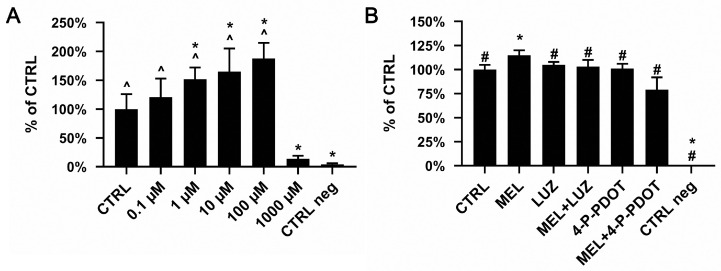
The influence of melatonin (**A**) and its inhibitor receptors (**B**) on ADSC viability. Each bar represents the mean percentage ± SD of the control cell viability (100%). (**A**) ADSC viability after treatment with melatonin at different concentrations: 0.1 μM, 1 μM, 10 μM, and 100 μM, expressed as a percent of CTRL. CTRL neg—cells treated with 1% of Triton X (ANOVA, post hoc Tukey, * *p* < 0.05 vs. CTRL, ^ *p* < 0.05 vs. 1000 µM, n = 8). (**B**) Viability of ADSCs grown for 48 h in the presence of melatonin (MEL), melatonin with luzindole (MEL + LUZ), melatonin with 4-P-PDOT (MEL + 4-P-PDOT), luzindole (LUZ), and 4-P-PDOT. CTRL—untreated cells; CTRL neg—cells treated with 1% of Triton X (ANOVA, post hoc Tukey, mean ± SD, * *p* < 0.05 vs. CTRL, # *p* < 0.05 vs. MEL, n = 6).

**Figure 2 pharmaceuticals-16-01236-f002:**
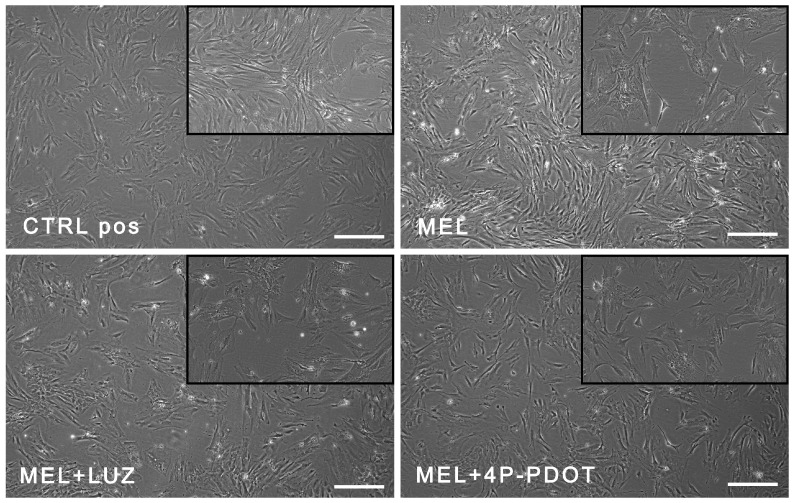
Unchanged morphology of ADSCs after a 48 h exposure to melatonin (MEL), melatonin with luzindole (MEL + LUZ), and melatonin with 4-P-PDOT (MEL + 4-P-PDOT), as compared to the untreated cells (CTRL). Scale bars—30 µm (magn. 40×; magn. in the miniature: 100×).

**Figure 3 pharmaceuticals-16-01236-f003:**
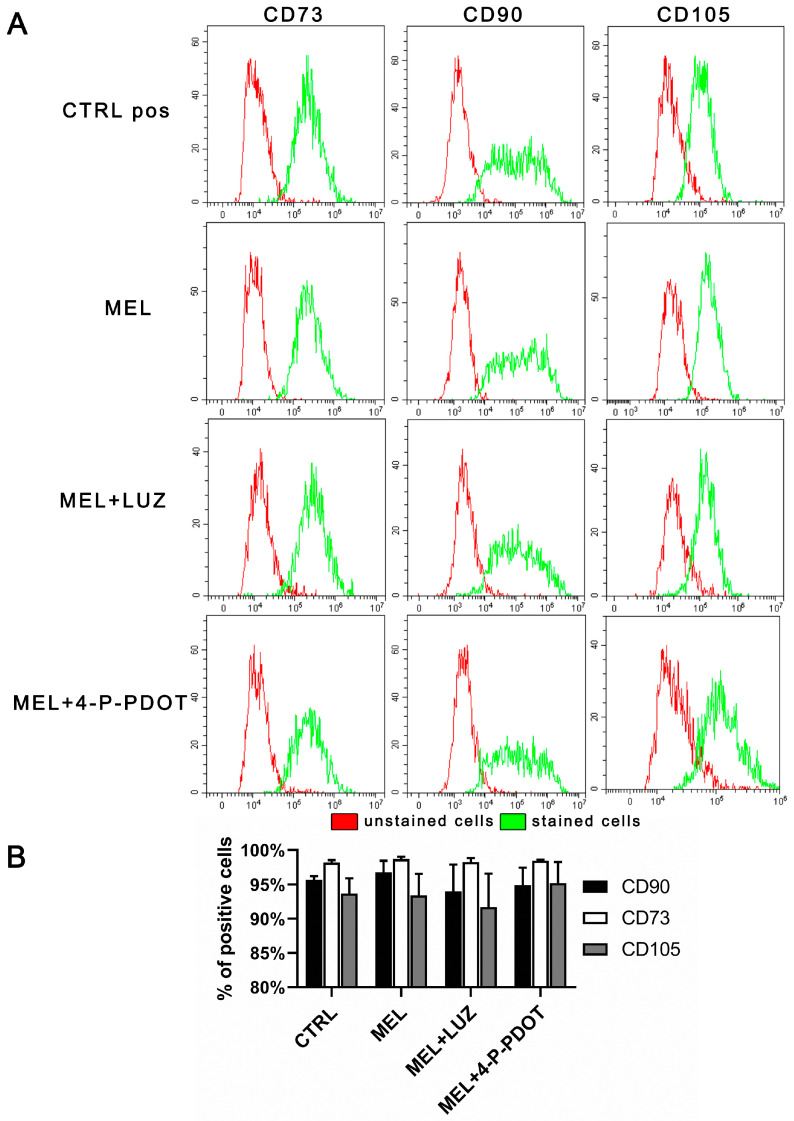
Expression of mesenchymal stem cell markers—CD73, CD90, and CD105—after a 48 h culture in the presence of melatonin (MEL), melatonin with luzindole (MEL + LUZ), melatonin with 4-P-PDOT (MEL + 4-P-PDOT), and in the control cells. (**A**) FACS histograms representative of CD73+, CD90+, and CD105+ ADSC cells. (**B**) Number of CD73+, CD90+, and CD105+ cells as a percentage of cultured ADSCs (ANOVA, post hoc Tukey, *p* < 0.05, mean ± SD, n = 3).

**Figure 4 pharmaceuticals-16-01236-f004:**
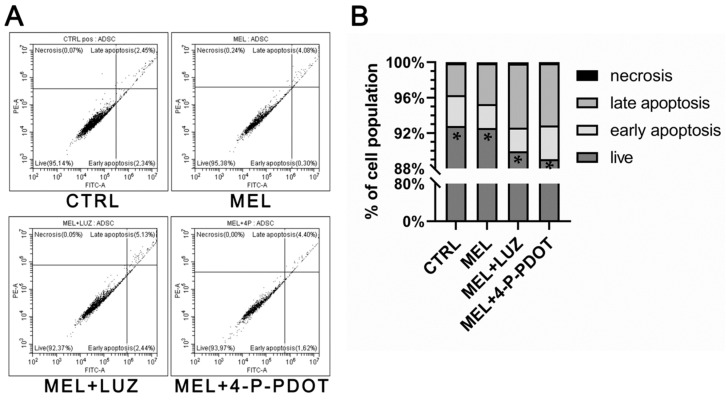
Detection of apoptotic or necrotic cells in cultures treated for 48 h with melatonin (MEL), melatonin with luzindole (MEL + LUZ), and melatonin with 4-P-PDOT (MEL + 4-P-PDOT). (**A**) FACS analyses representative of the experimental populations of ADSCs. (**B**) Mean percentage of live-, early-, or late-apoptotic and necrotic cells in cultured ADSC populations (* *p* < 0.05 as compared to necrotic and apoptotic cells, ANOVA, post hoc Tukey, *p* < 0.05, n = 3).

**Figure 5 pharmaceuticals-16-01236-f005:**
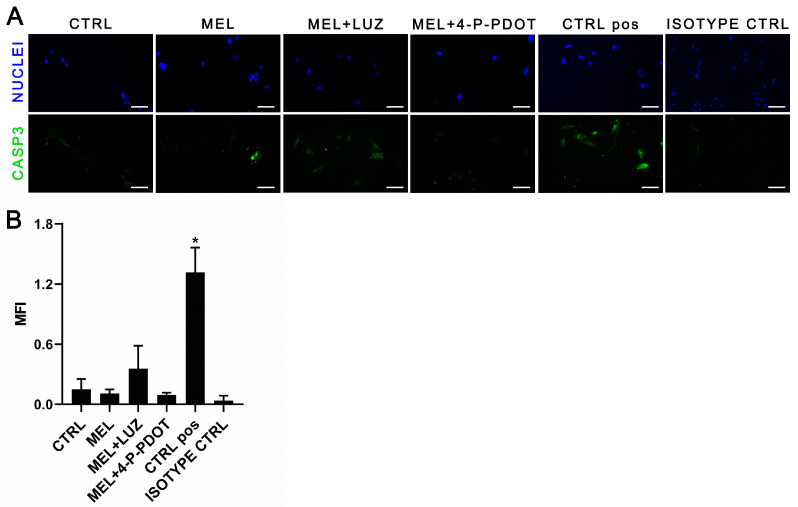
Immunodetection of cleaved caspase-3 in ADSCs treated for 48 h with melatonin (MEL), melatonin with luzindole (MEL + LUZ), and melatonin with 4-P-PDOT (MEL + 4-P-PDOT), compared to untreated cells (CTRL). (**A**) Polyclonal primary antibody (1:400) and secondary antibody conjugated with fluorochrome Alexa Fluor 488 (1:1000) were used for cleaved caspase-3 (Asp175) visualized with FITC (green). Cell nuclei were visualized with DAPI (blue). Apoptotic ADSCs cultured in 2% DMSO served as the positive control (CTRL pos). In experimental groups: scale bars—15 µm, magn. 100×. In isotype controls: scale bars—40 µm, magn. 40×. (**B**) Mean fluorescence intensity (MFI) measured with ImageJ software v. 1.52a (ANOVA, post hoc Tukey, * *p* < 0.05 vs. CTRL, mean ± SD, n = 3).

**Figure 6 pharmaceuticals-16-01236-f006:**
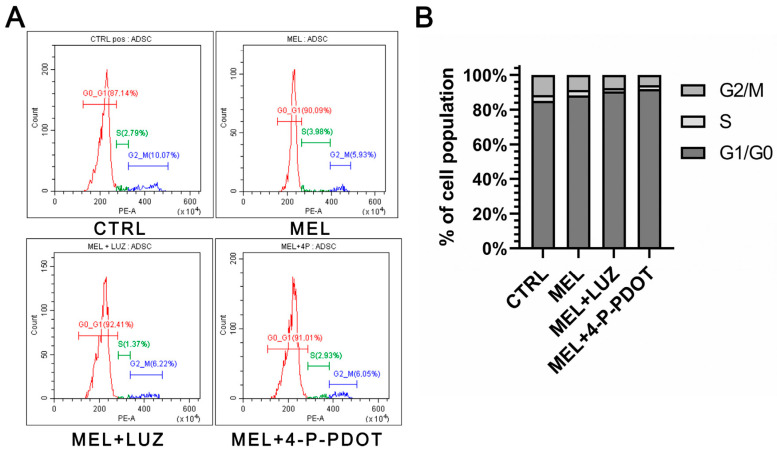
Flow cytometric analysis of cell cycle in ADSCs treated for 48 h with melatonin (MEL), melatonin with luzindole (MEL + LUZ), and melatonin with 4-P-PDOT (MEL + 4-P-PDOT), and in untreated (CTRL) cells. (**A**) FACS histograms representative of experimental populations of ADSCs in different phases of the cell cycle. (**B**) Mean percentage of ADSCs representing different phases of cell cycle in experimental cultures (ANOVA, post hoc Tukey, *p* < 0.05, n = 3).

**Figure 7 pharmaceuticals-16-01236-f007:**
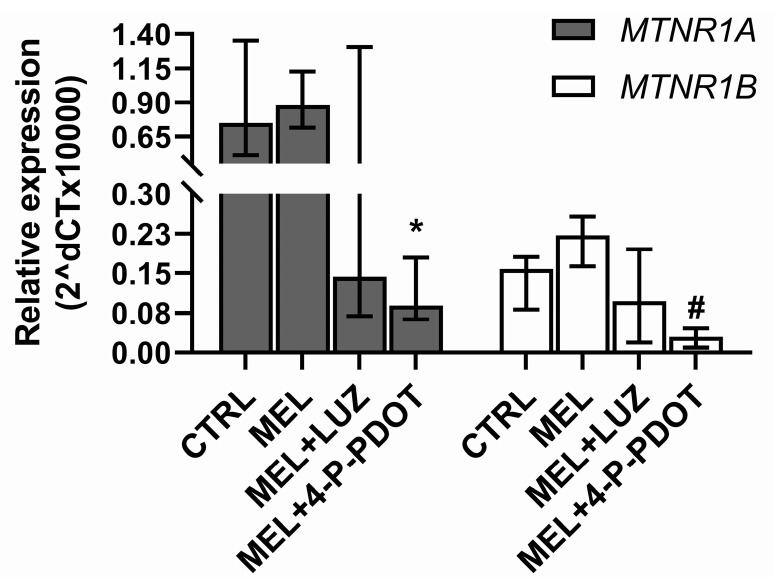
Expression of *MTNR1A* and *MTNR1B* genes in ADSCs after a 48 h exposure to: melatonin (MEL), melatonin with luzindole (MEL + LUZ), melatonin with 4-P-PDOT (MEL + 4-P-PDOT), and in control (CTRL) cells. Kruskal–Wallis test with multiple comparisons, medians, and quartiles (* *p* < 0.05 vs. CTRL; # *p* < 0.05 vs. MEL, n = 6).

**Figure 8 pharmaceuticals-16-01236-f008:**
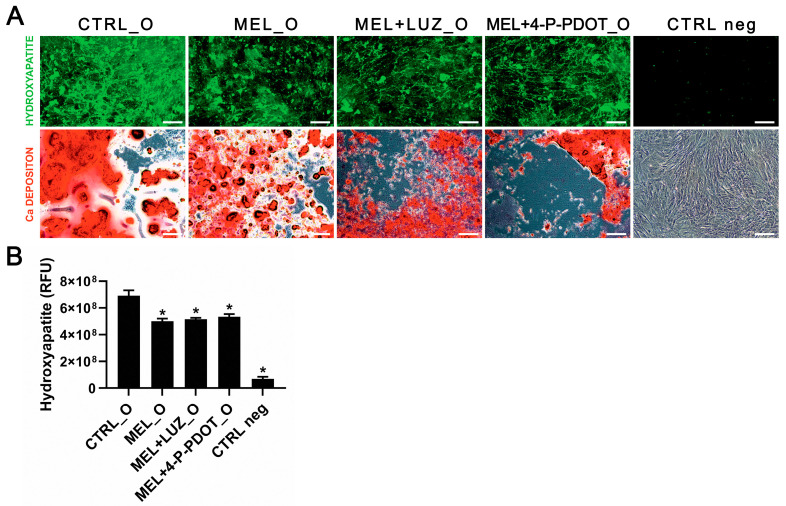
Effect of melatonin on hydroxyapatite and calcium deposition during ADSC differentiation. (**A**) The hydroxyapatite and calcium deposition secreted by ADSCs after 7 days of differentiation with melatonin (MEL_O), melatonin with luzindole (MEL + LUZ_O), and melatonin with 4-P-PDOT (MEL + 4-P-PDOT_O), as well as a culture of untreated, differentiated (CTRL_O), and undifferentiated ADSCs (CTRL neg). The process of extracellular calcium deposition was assessed using staining with Alizarin red solution (red color) and the identification of hydroxyapatite with an Osteoimage Kit (green fluorescence). (**B**) Relative fluorescence of hydroxyapatite in examined groups (means ± SD, * *p* < 0.05 vs. CTRL_O, ANOVA, post hoc Tukey, *p* < 0.05, means ± SD, n = 6). Scale bars—15 µm; magn. 100×.

**Figure 9 pharmaceuticals-16-01236-f009:**
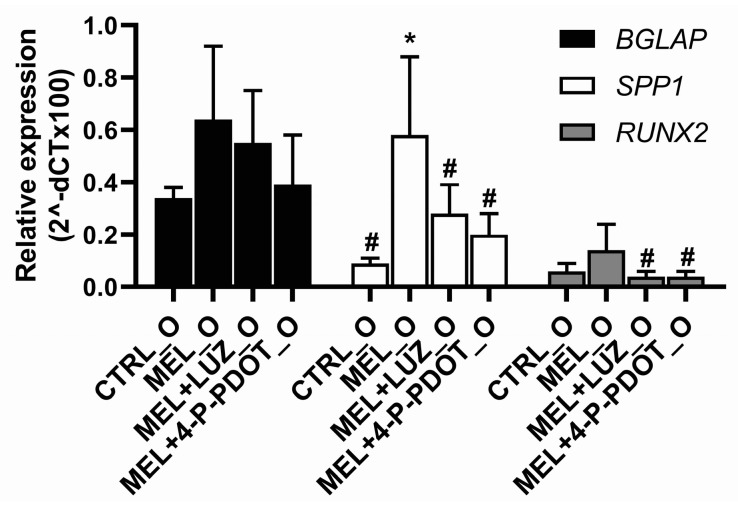
Expression of *BGLAP*, *SPP1*, and *RUNX2* genes in ADSCs after 7-day differentiation with melatonin (MEL_O), melatonin with luzindole (MEL + LUZ_O), melatonin with 4-P-PDOT (MEL + 4-P-PDOT_O), and in untreated cells (CTRL_O) (ANOVA, post hoc Tukey, *p* < 0.05, means ± SD, n = 6, * *p* < 0.05 vs. CTRL_O, # *p* < 0.05 vs. MEL_O).

**Figure 10 pharmaceuticals-16-01236-f010:**
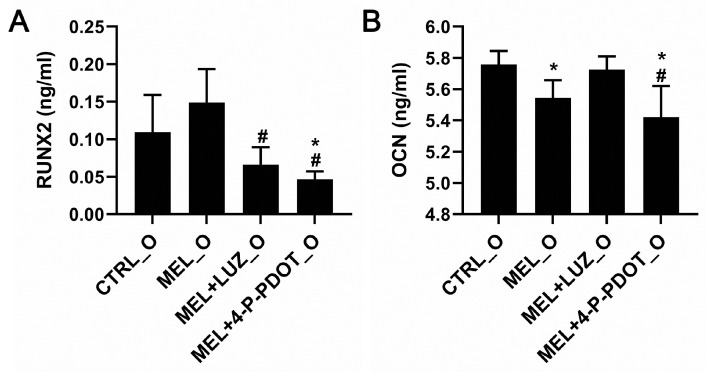
The concentrations of RUNX2 (**A**) and osteocalcin (**B**) in ADSCs after 7-day differentiation into osteoblasts. (**A**) * *p* < 0.05 vs. CTRL_O, # *p* < 0.05 vs. MEL_O, n = 6. (**B**) * *p* < 0.05 vs. CTRL_O; # *p* < 0.05 vs. MEL + LUZ_O, ANOVA, post hoc Tukey, *p* < 0.05, means ± SD, n = 6.

**Figure 11 pharmaceuticals-16-01236-f011:**
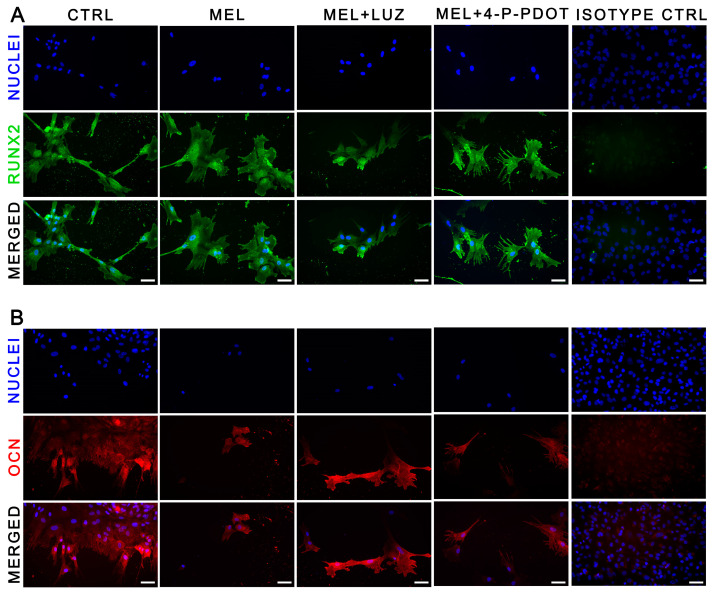
Visualization of RUNX2 (**A**) and osteocalcin (**B**) in differentiated ADSCs. (**A**) Monoclonal primary antibody (1:50) and secondary antibody conjugated with fluorochrome Alexa Fluor 488 (1:1000) were used for RUNX2 visualized with FITC (green). (**B**) Monoclonal primary antibody (1:500) and secondary antibody conjugated with fluorochrome Alexa Fluor 568 (1:1000) were used for OCN visualized with TRITC (red). Cell nuclei were visualized with DAPI (blue). Scale bars—15 µm, are representative for all images; magn. 100×.

## Data Availability

Data are contained within the article.

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
