# Peer review of "Regulation of Adipose-Derived Stem Cell Activity by Melatonin Receptors in Terms of Viability and Osteogenic Differentiation"

_pharmaceuticals, 2023, doi:10.3390/ph16091236_

Round 1

Reviewer 1 Report

The present study describes the effect of melatonin on osteogenesis from ADSC. The authors explain in a detailed manner that melatonin is not toxic to ADSC and does not interfere with ADSC osteogenic transdifferentiation. Although the manuscript is well presented, there are some points the authors should address before considering its publication:

In section 2.2, it is not clear the statement "... not toxic effect" the author should consider a more descriptive statemenr. 

In figure 1A the MTT results related to cells proliferation in Melatonin 100 uM  differ from others... the authors should explain this phenomenon

In Figure 2, the authors should use another microscope magnification in order to describe in a more detail the effects of melatonin   and its receptors for example, they are encourage to use a single cell magnification. 

In figure 3, the authors should also include analyses of MFI in conjunction with percentages

Figure 4 and 5, the authors should include positive apoptosis and necrosis controls to validate their methods 

Figure 5, the authors should support their findings with MFI analyses, a graphical comparison is welcome 

Figure 6, the authors should use cell cycle inhibitors to validate their findings 

Figure 8, it is expected that Mel antagonists revert Mel activity ... 

Figure 11, MFI quantification is welcome

Lines 309-310 "This may indicate that promotion of viability is regulated by the melatonin receptor pathway in ADSCs cultured in vitro." is mereley speculative

Lines 364-365 "We did not observe enhanced expression of cleaved caspase-3 at either mRNA or protein levels" This statement is incorrect because it is not supported by their findings, also cleaved caspase 3 it is not identified by mRNA...

Author Response

Thank you for recognizing the value of our work. Please find our answers below.

Reviewer 1

  1. In section 2.2, it is not clear the statement "... not toxic effect" the author should consider a more descriptive statement. 

As suggested, we have described it more precisely in this section.

  1. In figure 1A the MTT results related to cells proliferation in Melatonin 100 µM differ from others... the authors should explain this phenomenon

We expanded description of this results in 2.2 subsection addressing them to ATP assay. We hope that now the description is clearer.

  1. In Figure 2, the authors should use another microscope magnification in order to describe in a more detail the effects of melatonin   and its receptors for example, they are encouraging to use a single cell magnification. 

As suggested, In the figure 2 in the upper-right quadrant of each image we added the microphotography taken at higher magnification (x100) to better visualize the cell morphology. We also kindly inform that in the case of acceptation of this article, the high-resolution images will be available for interested readers at the online version of the manuscript.

  1. In figure 3, the authors should also include analyses of MFI in conjunction with percentages

Thank you for this suggestion. We have analyzed these results both by the percentage content of detected signals and MFI. We find the presentation of percentages of populations more informative and correct than the presentation of MFI. Calculating the percentage of detected signal is also related to the MFI and include itself the normalization of fluorescence signal. Therefore, joining the presentation of MFI data with percentages seems unnecessary. Moreover, the percentage presentation of expressed MSC markers is consistent with recommended minimal criteria for defining multipotent mesenchymal stem cells suggested by The International Society for Cellular Therapy. However, we respectfully provide the MFIs raw data below to help you evaluate this further:

Median Fluorescence Intensity

CD90

CD73

CD105

PC

UC

PC-UC

PC

UC

PC-UC

PC

UC

PC-UC

MEL

109821,2

1673,9

108147,3

233587,3

11477,5

222109,8

119317,3

16267,8

103049,5

MEL

102914

1545,1

101368,9

232820,8

10671,9

222148,9

237189,1

14909,7

222279,4

MEL

113302,8

1509,9

111792,9

215046,2

9981,2

205065

138003,3

14318,3

123685

CTRL

85456,1

8207,5

77248,6

202755

8197,8

194557,2

92342,6

10880,2

81462,4

CTRL

89576,5

1202,3

88374,2

196893,3

11133,5

185759,8

89105

12438,5

76666,5

CTRL

72034,3

1202,3

70832

226205,9

9698,7

216507,2

99928,9

13155,7

86773,2

MEL+LUZ

104082,2

1414,3

102667,9

202030,2

10239

191791,2

134573,7

14705,4

119868,3

MEL+LUZ

109548,8

1718,9

107829,9

218507,9

11502

207005,9

118056,4

17014,7

101041,7

MEL+LUZ

112332

1945,8

110386,2

273608,1

13064,7

260543,4

135650,4

49640,6

86009,8

MEL+4P-PDOT

298647,7

10056,3

288591,4

175077

10072,2

165004,8

115714,3

14427,2

101287,1

MEL+4P-PDOT

83051,1

1752,9

81298,2

282726,8

11488,7

271238,1

110295,3

16908,1

93387,2

MEL+4P-PDOT

135450,5

17451,6

117998,9

3278895,3

12001

3266894,3

133037,8

17310,2

115727,6

  1. Figure 4 and 5, the authors should include positive apoptosis and necrosis controls to validate their methods 

We agree that providing additional internal controls like the apoptosis or necrosis inducer would improve the quality of presented investigation data and also would made the applied test more credible. However, this assay is commonly used and the method is well known in scientific community. We believe that the principals of this test and properly set measurements are enough for presented data to be trustworthy. We also would like to assure that our internals standard operating procedures in the laboratory applies QC of instruments and every new introduced assay prior the investigating procedures. Furthermore, to confirm the specificity of annexin v/pi test (fig. 4) we actually performed reference analysis of the casp-3 detection (fig. 5) where the positive apoptosis control was provided indeed. The presented “CTRL pos” group in figure 5 are the cells treated with 2% of DMSO being well described apoptotic inducer, what is mentioned in the figure caption and 4.6.2 subsection.

  1. Figure 5, the authors should support their findings with MFI analyses, a graphical comparison is welcome 

Thank you for pointing this out. As suggested, we have provided the MFI analysis. We agree that this comparison underlines that what is visible in the microphotographs and makes it more readable.

  1. Figure 6, the authors should use cell cycle inhibitors to validate their findings 

Regarding this valuable remark we would like to refer to our answer to comment nr 5. We understand that providing additional internal controls would increase the value of applied test, however like the annexin v/pi test in this case the used PI/RNase staining solution is well recognized method in scientific community. We believe that obtained results are trustworthy and providing cell cycle inhibitors should be used in method standardization what we did prior the experiment.

  1. Figure 8, it is expected that Mel antagonists revert Mel activity ... 

In this assessment we aimed to verify if the melatonin alters the osteogenesis. The osteogenic differentiation is considered successful when hydroxyapatite or calcium shows up. The presented quantitative analysis in this case is not truly significant for our aim of the study, because the osteogenesis is a time-related process and such differences in total amount of hydroxyapatite between culture dishes may be expected. We decided to show the quantitative differences because it may be informative for planning further investigation. The lowered level of hydroxyapatite in study groups shows that the addition of melatonin probably slows down the process of mineralization. However, to confirm this, the study should be extended with the same analysis but performed in a few time points. As observed the MEL antagonists are expected to revert its visible effect, however it was not noticed in this assessment. For better explanation of this findings, we edited the discussion and provided a new commentary on these results.

  1. Figure 11, MFI quantification is welcome

Thank you for this suggestion. We do not find the need for quantitative comparison of these results. This assessment was to confirm whether cells express osteogenic markers in differentiated ADSC in study groups. In this case the immunodetection of osteogenic markers and simple yes or no analysis gave us a satisfactory answer. The presence of RUNX2 or OCN proves that the osteogenesis was not disturbed.

  1. Lines 309-310 "This may indicate that promotion of viability is regulated by the melatonin receptor pathway in ADSCs cultured in vitro." is mereley speculative

Thank you for this valuable observation. We edited the discussion to make it clear and consistent with our thoughts, and we provided a new commentary on these results.

  1. Lines 364-365 "We did not observe enhanced expression of cleaved caspase-3 at either mRNA or protein levels" This statement is incorrect because it is not supported by their findings, also cleaved caspase 3 it is not identified by mRNA...

The caspase 3 was assessed at protein level and RNA level. This is described in 2.4. titled “Effects of melatonin on ADSC apoptosis and necrosis” and 2.6 titled “Effects of melatonin on apoptotic (CASP3, CASP7) and melatonin receptors (MEL1a, MEL1b) gene expression” subsections. We highlighted this sentence with red in revised version. It has been stated: “(...)No apoptosis-specific expression of CASP3 and CASP7 was detected (data not shown).”

Reviewer 2 Report

The main aim of the study is to evaluate the effects of melatonin on the functional potential of mesenchymal stem cells in the vascular fraction of adipose tissue.

What is currently known about the effects of melatonin on mesenchymal stem cells?

1) Melatonin is involved in processes regulating the vital activity (proliferation, survival, programming, maintenance of stemness) of different types of stem cells. It is known that melatonin can reduce ferroptosis of medullary mesenchymal stem cells through PI3K/AKT/mTOR signalling pathways [Li M, Yang N, Hao L, Zhou W, Li L, Liu L, Yang F, Xu L, Yao G, Zhu C, Xu W, Fang S. Melatonin inhibits the ferroptosis pathway in rat bone marrow mesenchymal stem cells by activating the PI3K/AKT/mTOR signalling axis to attenuate steroid-induced osteoporosis. Oxid Med Cell Longev. 2022 Aug 18;2022:8223737. doi: 10.1155/202222/8223737], i.e. melatonin has points of influence on mesenchymal stem cells that lead to changes in their reactivity to microenvironmental factors. 2) Melatonin promotes adipogenic differentiation of mesenchymal stem cells independent of the involvement of the Wnt pathway, and chondrogenic and osteogenic differentiation with its involvement [Hardeland R. Melatonin and the Programming of Stem Cells. Int J Mol Sci. 2022 Feb 10;23(4):1971. doi: 10.3390/ijms23041971]. The ability of melatonin to exert antioxidant effects and to participate in the regulation of osteogenic differentiation has been demonstrated. Treatment of gingival mesenchymal stem cells with melatonin promoted the activation of alkaline phosphatase and cell mineralisation, and abolished the pro-inflammatory effect of hydrogen peroxide on the cells [Solá VM, Aguilar JJ, Farías A, Vazquez Mosquera AP, Peralta López ME, Carpentieri AR. Melatonin protects gingival mesenchymal stem cells and promotes their differentiation into osteoblasts. Cell Biochem Funct. 2022 Aug;40(6):636-646. doi: 10.1002/cbf.3733]. Melatonin enhances osteogenesis (increased expression of ALP, OCN, runt-related transcription factor 2, and osterix) and angiogenesis (increased expression of VEGF, angiopoietin-2, and angiopoietin-4) of mesenchymal stem cells in vitro [Zheng S, Zhou C, Yang H, Li J, Feng Z, Liao L, Li Y et al. Melatonin accelerates repair of osteoporotic bone defects by promoting osteogenesis-angiogenesis coupling. Front Endocrinol (Lausanne). 2022 Feb 22;13:826660. doi: 10.3389/fendo.2022.826660]. 3) The authors conducted a comparative study of the effect of preconditioning of mesenchymal stem cells from bone marrow and from the vascular fraction of rat adipose tissue and found no increase in cell survival, suppression of BAX expression and increased expression of BCL2, melatonin receptors 1 and 2. Osteocalcin expression was higher in medullary mesenchymal stem cells and calcium deposition was greater in mesenchymal table cells of the vascular fraction of adipose tissue [Rafat A, Mohammadi Roushandeh A, Alizadeh A, Hashemi-Firouzi N, Golipoor Z. Comparison of The Melatonin Preconditioning Efficacy between Bone Marrow and Adipose-Derived Mesenchymal Stem Cells. Cell J. 2019 Jan;20(4):450-458. doi: 10.22074/cellj.2019.5507].

Underlying the enhancement of osteogenic differentiation of mesenchymal stem cells is signalling through the melatonin-2 receptor and the involvement of the MEK/ERK (1/2) pathway [Radio NM, Doctor JS, Witt-Enderby PA. Melatonin enhances alkaline phosphatase activity in differentiating human adult mesenchymal stem cells grown in osteogenic medium via MT2 melatonin receptors and the MEK/ERK (1/2) signalling cascade. J Pineal Res. 2006 May;40(4):332-42. doi: 10.1111/j.1600-079X.2006.00318.x].  4) The presence of melatonin at a physiological dose of 20-200 pg/ml in the osteogenic medium when culturing mesenchymal stem cells from bone marrow or vascular fraction of adipose tissue of rats showed a greater osteogenic differentiation of cells from bone marrow (alkaline phosphatase), also showed differences in mineralisation, the number of cells in apoptosis is higher in adipose tissue and decreased viability [Zaminy A, Ragerdi Kashani I, Barbarestani M, Hedayatpour A, Mahmoudi R, Farzaneh Nejad A. Osteogenic differentiation of rat adipose tissue-derived mesenchymal stem cells in comparison with bone marrow-derived mesenchymal stem cells: melatonin as a differentiation factor. Iran Biomed J. 2008 Jul;12(3):133-41].

The study carried out by the authors is in line with the search for the application of mesenchymal stem cells with high osteogenic potential for regenerative medicine, but is not distinguished by originality.

This work shows the effect of micromolar doses of melatonin during prolonged exposure of adipose-derived mesenchymal stem cells on the functional properties of the cells, which can be considered novel.

The conclusions are consistent with the data presented in this paper.

The literature used to discuss our own research is current and sufficient in number.

Performance questions:

1. The logic of assessing cell viability by different methods is not clear - the MTT test is based on studying the intracellular activity of NADPH-dependent oxidoreductases, which indirectly indicates the metabolic activity/proliferation of cells, and the ATP assay reflects the activity of mitochondria. In the second case, the authors wanted to show that the energy balance is not disturbed, but if an MTT reaction takes place, it is only possible in a living cell.

2. In describing the phases of the cell cycle, the authors directly ignore the mitotic phase, which, when samples are analysed on a flow cytometer, is common to the G2 and M regions and is commonly referred to as G2/M. In addition, in the description of the phases of the cell cycle, the authors directly ignore the mitotic phase, which is common to the G2 and M regions and is commonly referred to as G2/M. It is also mentioned in the description of the methodology and in Figure 6, but not in the text.

Author Response

Thank you for recognizing the value of our work. Please find our answers below.

Reviewer 2

  1. “What is currently known about the effects of melatonin on mesenchymal stem cells?(…)”

Regarding the asked question and introducing a brief review of the current knowledge on melatonin activity we would like to thank for recognizing the properties of this compound and raising a few references that support our hypothesis. We decided to cite some of suggested sources in the improved parts of the text.

  1. The logic of assessing cell viability by different methods is not clear - the MTT test is based on studying the intracellular activity of NADPH-dependent oxidoreductases, which indirectly indicates the metabolic activity/proliferation of cells, and the ATP assay reflects the activity of mitochondria. In the second case, the authors wanted to show that the energy balance is not disturbed, but if an MTT reaction takes place, it is only possible in a living cell.

MTT was used for the evaluation of toxic dose of melatonin. We wanted to establish the possibly highest nontoxic dose of melatonin, but the one which shows any metabolic response. The dose of 100 µM was the cut-off point for analyzed concentrations. The tested range of doses was chosen based on the previously reported concentrations by several authors. After establishing the dose, we wanted to verify the exact activity of the melatonin and its inhibitors on tested cells. We decided to use the ATP assay due to its higher sensitivity and its non-questionable reflection of the mitochondria activity which occurs only in living cells. The assessment showed that none of the tested bioactive compounds kill the cells, thus we could proceed further evaluation. To clarify the case of using both assays we added a brief summary in 2.2. subsection.

  1. In describing the phases of the cell cycle, the authors directly ignore the mitotic phase, which, when samples are analyzed on a flow cytometer, is common to the G2 and M regions and is commonly referred to as G2/M. In addition, in the description of the phases of the cell cycle, the authors directly ignore the mitotic phase, which is common to the G2 and M regions and is commonly referred to as G2/M. It is also mentioned in the description of the methodology and in Figure 6, but not in the text.

We edited the 2.5 subsection and added information about G2/M in text. We also summarized this subsection with a sentence “We did not observe significant differences between groups in G2/M phase specific for growth and division of cells, it proved that melatonin did not change the cell division activity.. Thank you for your comment.

Reviewer 3 Report

Based on the potential of this paper to contribute to the field of regenerative medicine, I recommend revision to address the missing points and enhance the paper's scientific quality. With appropriate improvements, this study could provide valuable insights into the use of melatonin and ADSCs for bone regeneration and be suitable for publication in Pharmaceutics.

1- The introduction could strengthen the rationale by providing more comprehensive statistics on osteosarcoma prevalence and its treatment limitations.

2- Detailed explanations of experimental methods, statistical analyses, and individual results for each experiment are needed.

3- Address potential experimental limitations and discuss the reasons for differences in gene expression between melatonin and its inhibitors.

4- Clarify the number of replicates and statistical analyses conducted.

Discuss control measures and assay validations to ensure result validity.

5- Address ethical considerations related to stem cells and experimental animals if applicable.

6- Explain mechanistic insights into how melatonin affects gene expression, cell viability, and differentiation.

7- Suggest avenues for further research, such as long-term effects or synergies with other compounds.

The article requires significant revisions to enhance its clarity, coherence, and overall quality. There are instances where the content lacks smooth transitions and is fragmented, making it challenging to follow.
Simplifying sentence structures and using transitional phrases would greatly improve the flow and readability of the text. Additionally, attention to grammar, punctuation, and academic writing conventions is necessary. Clear and well-defined objectives and hypotheses should be stated in the introduction, while the results need to be presented and discussed in an organized manner.

The conclusion should succinctly summarize the key findings without introducing new information. Given the extent of these improvements, a major revision is warranted to bring the manuscript up to a high academic standard. 

The article requires significant revisions to enhance its clarity, coherence, and overall quality. 

Author Response

Thank you for recognizing the value of our work. Please find our answers below.

Reviewer 3

  1. The introduction could strengthen the rationale by providing more comprehensive statistics on osteosarcoma prevalence and its treatment limitations.

As suggested, we have added information on the epidemiology of osteosarcoma in the introduction.

  1. Detailed explanations of experimental methods, statistical analyses, and individual results for each experiment are needed.

We have extended the description of statistical analysis in sections of material and methods and results.

  1. Address potential experimental limitations and discuss the reasons for differences in gene expression between melatonin and its inhibitors.

As suggested, we briefly discussed the experimental limitations and the mentioned issue in discussion.

  1. Clarify the number of replicates and statistical analyses conducted. Discuss control measures and assay validations to ensure result validity.

We added the description of statistical analysis tests in sections of material and methods and results, and we added the information on replications in the figure captions.

  1. Address ethical considerations related to stem cells and experimental animals if applicable.

We find this issue inapplicable in this case, however we clarified the source of cell line in the material and methods section. All experiments were conducted in vitro on standardized cell line from ATCC cell bank. The cells were obtained commercially after signing an appropriate agreement with its distributor and no further approval by local ethical committee is needed to perform experiments. The used cells were primary, adult (somatic) stem cells obtained from patients under informed consent which was obtained by ATCC.

  1. Explain mechanistic insights into how melatonin affects gene expression, cell viability, and differentiation.

Thank you for this suggestion. We addressed this briefly in the discussion.

  1. Suggest avenues for further research, such as long-term effects or synergies with other compounds.

As suggested, we added our thoughts on research perspectives at the end of discussion.

  1. The article requires significant revisions to enhance its clarity, coherence, and overall quality. There are instances where the content lacks smooth transitions and is fragmented, making it challenging to follow. Simplifying sentence structures and using transitional phrases would greatly improve the flow and readability of the text. Additionally, attention to grammar, punctuation, and academic writing conventions is necessary. Clear and well-defined objectives and hypotheses should be stated in the introduction, while the results need to be presented and discussed in an organized manner.

We agree with your comment, and we apologize for any inconvenience that may be encountered while reading this article. In addition to the scientific goal of the research, we also aimed to describe it as clearly as possible and did our best to make it straight and understandable. Before submitting the manuscript, we sent it to a professional language editor who made grammatical corrections. We understand that any text can be written better and more accurately. We believe that this manuscript in its current form is readable for potential readers, so we have decided to improve the substantive message but not to apply extensive language editing. We kindly inform that we have improved the discussion, clarified the aim of the study, and drawn clear conclusions.

  1. The conclusion should succinctly summarize the key findings without introducing new information.

We provided new conclusions summarizing the key findings and removed irrelevant statements.

Round 2

Reviewer 1 Report

The authors made an extraordinary effort to answer satisfactory all the observations. 

Reviewer 3 Report

Accepted